# Efficacy of Imidacloprid Seed Treatments against Four Wheat Aphids under Laboratory and Field Conditions

**DOI:** 10.3390/plants12020238

**Published:** 2023-01-04

**Authors:** Zhi Zhang, Yaping Li, Xiangrui Li, Xun Zhu, Yunhui Zhang

**Affiliations:** 1State Key Laboratory for Biology of Plant Diseases and Insect Pests, Institute of Plant Protection, Chinese Academy of Agricultural Sciences, Beijing 100193, China; 2Beijing Plant Protection Station, Beijing 100029, China

**Keywords:** wheat aphid, imidacloprid seed treatment, *Sitobion avenae* (Fabricius), *Rhopalosiphum padi* (Linnaeus), *Schizaphis graminum* (Rondani), *Metopolophium dirhodum* (Walker)

## Abstract

Imidacloprid seed treatments are effective at reducing the cohorts of many insect pests on crops such as cotton, corn, and cereals. The effects of imidacloprid seed treatments depend on the aphid species. In China, there are four wheat aphid species—*Sitobion avenae* (Fabricius), *Rhopalosiphum padi* (Linnaeus), *Schizaphis graminum* (Rondani), and *Metopolophium dirhodum* (Walker)—and for a given region, these four aphid species differ in dominance with changes in cultivation practices and climate. Therefore, it is necessary to evaluate the effects of imidacloprid seed treatments on the four different aphid species. In experiments in the laboratory, imidacloprid seed treatments significantly reduced the survival rates of *S. avenae*, *R. padi*, and *S. graminum* to 57.33 ± 2.86%, 12.67 ± 1.92%, and 20.66 ± 2.33%, respectively, but for *M. dirhodum*, there was no significant difference between the control (96.33 ± 1.08%) and the treatment (97.00 ± 0.98%). The fecundities of the four aphid species were much reduced, especially for *R. padi* when feeding on treated wheat plants. For the field survey, only three aphid species were considered because the density of *S. graminum* was too low to be analyzed. The effects of imidacloprid seed treatment on the three aphid species in the field were consistent with the laboratory results. Imidacloprid seed treatment reduced the population sizes of *S. avenae* and *R. padi* at rates of 70.30 ± 3.15% and 87.62 ± 2.28%, respectively, for the whole wheat season in the field. For *M. dirhodum*, imidacloprid seed treatments were less effective, and the densities of *M. dirhodum* increased on four sample days. From this study, we confirmed that the effect of imidacloprid seed treatment varied with the composition of aphid species, being especially less effective for *M. dirhodum*.

## 1. Introduction

Wheat (*Triticum aestivum* L.) is one of the most important cereal crops and covers about 24 million ha in China, with a total output of about 135 million tons. Throughout its development, wheat is subject to various forms of damage by insect pests, including wheat aphids, wheat midges, and soil pests, which threaten wheat yield and quality. In China, aphids were not the main insect pests of wheat before the 1970s [1,2], but with changes in tillage [3,4,5], farming systems, and variety replacement [6], wheat aphids have become the most important insect pests, with a total acreage of up to 14 million ha in 2018 [7,8]. Wheat aphids feed on the functional blades and spikes from the booting stage to the milk development stage. In years with heavy aphid outbreaks, honey-dew excretions covering the surface of blades affect the respiration and photosynthesis of the plants, which results in significant yield loss [9]. According to the reports, an average of 500 aphids per 100 tillers is the threshold of wheat aphid infestation when management measures are to be taken [1]. In addition, wheat aphids are important insect vectors that carry and spread plant viruses between crops, especially the barley yellow dwarf virus (BYDY) [10,11,12,13]. In China, there are four wheat aphid species: *Sitobion avenae* (Fabricius), *Rhopalosiphum padi* (Linnaeus), *Schizaphis graminum* (Rondani), and *Metopolophium dirhodum* (Walker). In different regions, the dominant species differ. *S. avenae* is widespread in China and becomes dominant during the booting stage. *R. padi* is vigorous in humid regions. *S. graminum* has been recorded in most provinces of China but does heavy damage only in northern and northwestern China, where drought is common. *M. dirhodum* is relatively endemic and has been recorded in the middle and north of China, with heavy infestations in some provinces [9].

Wheat aphid control has been primarily dependent on foliar insecticides such as dimethoate, chlorpyrifos, pirimicarb, imidacloprid, and cyhalothrin since the 1990s [9,14,15]. The use of these insecticides has brought about a series of issues, such as resistance, environmental pollution, and natural enemy reduction [16,17]. It is therefore necessary to develop a better management strategy to control wheat aphids. Seed treatments in cotton, corn, cereal, sugar beet, and oilseed rape with neonicotinoid compounds such as imidacloprid, thiamethoxam, dinotefuran, and thiacloprid are effective for the control of sucking insect pests such as aphids, whiteflies, plant hoppers, and thrips; chewing pests, including some lepidopteran pests; and a number of coleopteran pests [18,19,20,21,22,23,24]. Among those seed treatment ingredients, imidacloprid has the longest history (since 1991) of improving crop protection against sucking insect pests because of its broad insecticidal spectrum, systemic action, and low residue [18,20,25]. Imidacloprid seed treatment was observed to reduce pest density on maize, such as that of wireworms, cutworms, aphids, leafhoppers, and frit fly (*Oscinella frit* L.) [26,27]. Imidacloprid seed treatment could provide long-term protection for wheat cultivars or barley against wheat aphids and thereby increase production [18,19,28]. In China, imidacloprid seed treatment could control aphids for the whole growing season of winter wheat [21]. Because of its effectiveness and convenience in the control of wheat aphids, imidacloprid seed treatment has been widely used in China. However, for aphid pests, Pons and Albajes (2002) found that the effectiveness of seed treatments varies with the aphid species on maize at the beginning of August, and the duration of the effectiveness also depends on the composition of aphid species [26]. To date, published reports studied wheat aphids as a mixed population when testing the effectiveness of imidacloprid seed treatment [17,21], and there has been no report on its effectiveness on specific wheat aphid species. In this article, we hypothesized that the effectiveness of imidacloprid seed treatment varies with different aphid species, and we used laboratory experiments and field surveys to determine quantitative susceptibility to imidacloprid seed treatment in four wheat aphid species. The results could provide guidance in making control decisions regarding wheat aphids using imidacloprid seed treatment.

## 2. Results

### 2.1. Laboratory Survival

Wheat aphids were placed on wheat blades on the thirtieth day after germination, and 5 days later, the survival rates of all aphid species feeding on untreated wheat tillers were significantly higher than those on treated wheat tillers, except for *M. dirhodum* (Figure 1). The survival rate of *M. dirhodum* on the treated seedlings (97.00 ± 0.98%) was slightly higher than that on the untreated seedlings (96.33 ± 1.08%), but the difference was not significant by paired bootstrap test. Survival rates on the other treated seedlings were otherwise significantly different from those on untreated seedlings, and *R. padi* had the lowest survival (12.67 ± 1.92%).

### 2.2. Laboratory Fecundity

Fecundity varied with the aphid species on control plots and was significantly different across the species (Figure 2). *R. padi* had the highest fecundity, and *S. graminum* had the second-highest fecundity: one adult of each species could lay 7.97 ± 0.187 offspring and 7.27 ± 0.11 offspring, respectively, in five days. *S. avenae* and *M. dirhodum* had similar lower fecundity on control plots and produced 6.86 ± 0.09 offspring and 6.86 ± 0.09 offspring, respectively. Imidacloprid seed treatment was able to reduce the fecundity of the four aphid species significantly compared with control plots (Figure 2). *R. padi* produced no offspring in five days in the test. *M. dirhodum* had the highest fecundity (4.17 ± 0.10)—significantly more than that of *S. avenae* (0.15 ± 0.00) and *S. graminum* (0.05 ± 0.01).

### 2.3. Field Population Dynamics

There were four wheat aphid species located in Langfang—*S. avenae*, *R. padi*, *S. graminum*, and *M. dirhodum*—but only three aphid species were considered because the density of *S. graminum* was too low to be analyzed. The population sizes of wheat aphids reached peaks around the middle of May every year during the wheat grouting stage (Figure 3). In the control plots, aphid populations were established with alates that readily produced young nymphs from the beginning of April, whereas in treated plots, there were exclusively alates, which reproduced much less. The population sizes of wheat aphids differed greatly across the years because of the weather conditions and the density of natural enemies. The wheat fields were less infested by wheat aphids in 2015 and 2016 but were heavily infested in 2017. Regarding the total aphid population sizes of these three years, *S. avenae*, accounting for 73.94% in control plots and 60.27% in treated plots, was the dominant species in this area, followed by *R. padi* and *M. dirhodum*. However, in 2016, the percentage of *M. dirhodum* reached 37.69% in control plots and 62.30% in treated plots.

In 2015, the aphid population sizes reached their peak on about the 13th to 17th of May. During the peak period, the highest total aphid population size per 100 tillers was 1164.33 ± 91.94 in the control, while in the treated plots this was significantly reduced to 383.5 ± 39.38. The effect of imidacloprid seed treatments varied with the specific aphid species. During the peak periods, imidacloprid seed treatments could significantly reduce the population sizes, and the control efficiency rates of *S. avenae* and *R. padi* were 47.06~80.33% and 58.95~85.75%, respectively. However, imidacloprid seed treatments had no significant effect on *M*. *dirhodum*. On the sample days of the 3rd, 13th, and 17th of May, population sizes of *M*. *dirhodum* in treated plots were higher than those in control plots at rates of 35.55~42.85% (Figure 3).

In 2016, the occurrence of wheat aphids in the wheat fields was similar to that in 2015, but the proportions of *R. padi* and *M*. *dirhodum* in total were 17.57% and 37.69%, greater than those of the previous year. The population sizes of *S. avenae* during the peak periods in control plots were much lower compared with those in 2015, with about a 58.8% reduction; the highest population size of *S. avenae* was only 409.67 ± 44.41 individuals per 100 tillers. However, the highest population sizes of *R. padi* (207.67 ± 41.15 individuals/100 tillers) and *M*. *dirhodum* (478.50 ± 268.82 individuals/100 tillers) were much larger than those in 2015 (112.16 ± 18.18 and 50.17 ± 15.77). During the peak periods, imidacloprid seed treatments could also significantly reduce the total aphid population by mainly reducing *S. avenae* and *R. padi*, at rates of 55.13~77.67% and 82.86~97.16%, respectively. For *M*. *dirhodum*, the effect of imidacloprid seed treatments was not significant on most survey days, except on the 23rd of April; unexpectedly, imidacloprid seed treatments increased *M*. *dirhodum*’s density on the 22nd of May and the 3rd of June, when the population sizes in the treated plots were higher than those in the control plots (Figure 3).

In contrast to the previous two years, in 2017, *S. avenae*, *R. padi*, and *M*. *dirhodum* developed simultaneously in wheat fields during a very short period, and peak dates when the highest population size appeared were delayed by about 1 week. The proportion of *S. avenae* was up to 76.68%, followed by 14.20% for *R. padi* and 9.11% for *M. dirhodum*. In control plots, the total population sizes increased abruptly up to 9448.33 ± 647.26 individuals per 100 tillers, including 8233.67 ± 589.11 *S. avenae*, 782.33 ± 157.66 *R. padi*, and 432.33 ± 104.99 *M. dirhodum*, whereas imidacloprid seed treatments greatly reduced the infections of *S. avenae* and *R. padi* at rates of 62.07~90.56% and 79.16~98.31%, respectively. The highest total population size of aphids during the peak period was lowered to 1179.00 ± 155.77 individuals per 100 tillers, and the highest population sizes of *S. avenae* and *R. padi* throughout the season were 777.33 ± 137.61 and 29.97 ± 18.99 individuals per 100 tillers. However, for *M. dirhodum*, the population density was not noticeably reduced, and on the 17th of April, its population slightly increased (Figure 3).

Imidacloprid seed treatments had significant control effects on *S. avenae* and *R. padi* throughout the seasons from 2015 to 2017. The population sizes of *S. avenae* combined with *R. padi* in treated plots throughout the season were up to 787.33 ± 139.67 aphids per 100 tillers—nearly 300 more than the control level (500 aphids per 100 tillers)—especially in the heavy occurrence year of 2017. In 2016, the density of aphids in treated plots was distinct from the control measurement, with the highest density of 372.83 ± 65.34 aphids on the 11th of May, which was below the control level of wheat aphids in China. Unfortunately, imidacloprid seed treatments failed to reduce the density of *M. dirhodum* in any of the tested years. During the peak periods of all examined years, the population densities of *M. dirhodum* were increased on four sample days.

## 3. Discussion

Imidacloprid seed treatment can protect young plants from attack by insect pests and is also better for environmental safety than seed treatments with other pesticides [10,20]. Because of these factors, imidacloprid seed treatments are widely used in cotton, corn, cereals, sugar beet, oilseed rape, and other crops [20]. It is now estimated that 60% of applications of neonicotinoid insecticides are delivered via soil or seed treatments [22]. Seed dressings with imidacloprid are effective at reducing many pest populations [22,29,30]. In addition, imidacloprid seed treatments have more systemic action than other insecticides, meaning that they are more effective for long-lasting control of insect pests in the early season of many plants [20]. For aphid pests, several papers have reported the effectiveness of imidacloprid in reducing aphid populations in winter cereals and sorghum [18,26,31,32,33]. Many efficacy trials of imidacloprid seed treatments took the aphid population as a mixed experimental object [17,21,31]. However, the effectiveness of seed treatments depends on the composition of aphid species [26]. Thus, in this paper, we evaluated the effectiveness of imidacloprid seed treatments on four wheat aphid species.

In our laboratory tests, imidacloprid seed treatment significantly reduced the survival and fecundity of *S. graminum*, *S. avenae*, and *R. padi* (Figure 1 and Figure 2). For *M. dirhodum*, the fecundity was significantly reduced, but the survival rate of the cohort feeding on treated plants was not significantly different from that of the cohort feeding on untreated plants. This lesser effectiveness was also confirmed by a life table analysis: most individuals of *M. dirhodum* feeding on treated plants grew to the adult stage and produced nymphs, but their development time was extended, and their lifespan and fecundity significantly declined [34]. These findings confirmed our preliminary efficacy data on the four aphid species collected from a wheat field in Beijing. However, caution should be taken in comparing laboratory tests with field monitoring as populations differ in sensitivity within China [7].

Previous studies showed that in field tests, seed treatment at an imidacloprid rate of 2.5 g/kg effectively reduced greenbug (*S. graminum*) abundance up to 79 days after treatment [32]. Some studies found that wheat seeds treated with imidacloprid were effective against wheat aphids throughout the wheat growing season [17,21]. In our field test, the population sizes differed greatly across the sampling years because the temperature and precipitation during the peak period in 2017 was more adaptable for wheat aphids, and maximum enemy–pest ratios in 2015 and 2016 were higher than that in 2017. We found that there was relatively stable effectiveness across the three years and the duration of effectiveness of imidacloprid seed treatment varied with the aphid species. Imidacloprid seed treatment significantly reduced the densities of *S. avenae* and *R. padi* from 2015 to 2017, and this effectiveness lasted for the whole wheat season. However, for *M. dirhodum,* the densities in treated plots were somehow higher than those in control plots. In previous reports, some increases were noted for *Ostrinia nubilalis* on treated plots of maize due to the moth preferring to lay eggs on the well-growing plants in the treated field [26]. According to previous studies, we deduced two possible reasons for the different effectiveness of imidacloprid seed treatment among the four wheat aphid species. First, imidacloprid has antifeedant action [29,30], and the susceptibility of aphid species to imidacloprid seed treatment is different. *M. dirhodum* has shown higher tolerance to most insecticides compared with the other aphid species [35]. Second, the differences in sensitivity of the four aphid species to imidacloprid seed treatment may be related to their habits and ecological niches on wheat, because *S. avenae* especially prefer the wheat ear, flag blades, sticks, and aggregates in the middle and upper parts of plants [36]. In Langfang, *M. dirhodum* prefers the lower parts of wheat plants. That would result in two impacts: a lower location would protect these aphids from being caught by natural enemies, and the control of *S. avenae and R. padi* would lead to weak interspecific competition and would broaden the ecological niche of *M. dirhodum*. This could result in an increase or lesser reduction of *M*. *dirhodum* in treated plots.

*M*. *dirhodum* is a cosmopolitan cereal pest [37], but in China, it is more common in northern areas [9]. In general, *S. avenae* is of greatest dominance in the most northern wheat fields. However, the dominance of aphid species could vary with changes in cultivation practice and the climate; for example, in 2016, the percentage of *M*. *dirhodum* reached 37.69% in control plots and 62.30% in treated plots. Thus, the practice of imidacloprid seed treatment should take the composition of aphid species into account and take more species-specific approach where the *M*. *dirhodum* is of high proportion [34].

In addition to the composition of aphid populations, we should consider many other possible factors when treating seeds with imidacloprid. First, it is necessary to carry out monitoring on the wheat aphids’ resistance to imidacloprid, due to some cohorts of *M. dirhodum* [35,38] and *R. padi* [34] having shown higher tolerance and moderate levels of resistance to imidacloprid and special ecological heritability [39]. Second, there must be consideration of weather conditions because soil moisture stress influences the translocation of insecticides [34,40,41]. When negative translocation of insecticides is observed, farmers should apply other measures at an appropriate time. Third, these compounds also carry risks to natural enemies and non-target areas, both during and after planting [25]. One study reported that imidacloprid seed treatments have no negative effects on Araneae, Coccinellidae, and Dermaptera, and only occasionally negatively affect Carabidae and Staphylinidae [10]. However, many studies found that imidacloprid and clothianidin seed treatments in the field significantly reduced the population densities of ladybirds, hoverflies, and parasitoids [17] and also have negative impacts on the health of honeybees [25,42]. In additional, imidacloprid seed treatments contaminate soil, water, and plant products, including pollen and nectar [25]. In the future, it is better to make the decision to treat wheat seeds based on an assessment of the risk of wheat aphid infestations.

## 4. Materials and Methods

### 4.1. Wheat Aphids

Laboratory colonies of wheat aphids comprising *S. avenae*, *R. padi*, *S. graminum*, and *M. dirhodum* were established from their field populations in Langfang (39.524° N, 116.685° E), Hebei province, 2010, by removing natural enemies over many generations. Thereafter, colonies of aphids were raised in the laboratory at the Institute of Plant Protection (Chinese Academy of Agricultural Science, Beijing, China) for bioassays. All lab aphid clones were maintained on wheat tillers in an environmental chamber at 20 ± 1 °C, at 60 ± 10% RH, and with a photoperiod of 16:8 h (light:dark).

### 4.2. Plant Material

Wheat seeds (Lunxuan 987) were purchased from the Fengda Seeds Co., Ltd. (Beijing, China).

### 4.3. Seed Treatment

Seed dressing was carried out by the Beijing branch of the Bayer company for laboratory and field experiments each year, and 2 mL of the formulated product (imidacloprid as the active ingredient, 600 g/L, high and low rates are 400 and 200 mL kg^−1^) diluted in 10 mL of water was used to treat each kg of seed. This low rate is the widely used recommend dose for wheat planting in China [21,33].

### 4.4. Laboratory Experiment

Ten dressed seeds or untreated seeds were planted in each plastic flowerpot (height 12 cm, upper and lower diameters 15.5 and 8 cm, thickness 0.5 cm). When wheat tillers had 3~5 blades, transparent polyethylene ecological boxes (length, width, and height 2.8 × 2.8 × 2.6 cm^3^; thickness 0.1 cm) were nipped on the leaves [42]. Each ecological box had a window formed with copper wire netting on the upper side of the box to avoid aphid escape and to also make them suitable for feeding aphids. The ecological boxes could be opened from the middle, and the edges were equipped with soft material to avoid damage to the clamped blades; therefore, the ecological boxes were convenient for observing the survival conditions and counting the numbers of laid nymphs.

As wheat plants grew, on the thirtieth day after germination, five adult aphids were chosen from the laboratory wheat aphid clones and put into each ecological box. The captive wheat aphids fed on the wheat blade clamped in the middle of the ecological box. The survival and fecundity of the captive wheat aphids were observed every 24 h for five days. All newborn nymphs from a numbered ecological box were transported into a dish bearing the same number, loaded with untreated wheat tillers, where the survival of the newborn nymphs was checked until they died. In this experiment, three replicates of 100 aphids were used for each treatment and control.

### 4.5. Field Experiment

The field survey was carried out during three consecutive wheat growing seasons from 2015 to 2017 at Langfang station (a station of the Institute of Plant Protection, Chinese Academy of Agricultural Science), Hebei province (39.524° N, 116.685° E). Here, wheat cultivation has a long history, and a previous study confirmed large populations of wheat aphids, including *S. avenae*, *R. padi*, and *M. dirhodum* [43]. Dressed wheat seeds of Lunxuan 987 were sown using a conventional drill on the following dates: 29 October 2014, 14 October 2015, and 15 October 2016. Three treated plots accompanied by an untreated plot on each longer side were located in the same place every season; every plot was 1.8 m in width and 56 m in length. Over the whole season, no other insecticides, fungicides, or herbicides were used in order to avoid their effects on the aphid population. From June to October, the field was left fallow before the next season.

From the reviving period, surveys in the treated and control plots were conducted approximately weekly beginning on 15 April 2015, 17 April 2016, and 4 April 2017 and continuing until the middle maturity stage. Ten samplings were conducted uniformly in each plot. On each sampling day, the numbers of winged and wingless individuals of different species, including *R. padi*, *S. avenae*, and *M. dirhodum*, were counted by visually searching 100 tillers of each sampling in six plots when the aphid density was less than 200 aphids per 100 tillers and searching 20 tillers of each sampling in six plots when the aphid density was more than 200 aphids per 100 tillers [17,32]. When it was raining or when crops were being irrigated, the sampling was done in advance or postponed by about 1~2 days.

### 4.6. Data Analysis

A bootstrap technique was conducted using TWOSEX-MSChart (downloaded from the website http://140.120.197.173/Ecology/prod02.htm, accessed 14 November 2022) in this study with 100,000 resamplings to estimate the standard errors of the survival rate and fecundity of each treatment and the control, following references [44,45,46]. The difference between the two treatments was compared via the paired bootstrap test with the results of the above 100,000 bootstrap resamplings. In this paper, the peak periods of aphid populations refer to the dates when the total population sizes in the control plot were above 500 aphids per 100 tillers during the growing season. Data are presented as the mean ± S.E. All graphs were created using SigmaPlot v.14.0.

## 5. Conclusions

Seed treatment with imidacloprid is popular in large wheat production regions in China. This study reconfirmed that imidacloprid seed treatments are very effective, but the effectiveness varies with the composition of aphid species. Imidacloprid seed treatment reduced the populations of *S. avenae* and *R. padi* but not that of *M*. *dirhodum* in laboratory tests and field surveys. There is a major risk of pest damage in areas where *M*. *dirhodum* occur, because our results showed a lesser control effect of imidacloprid seed treatment on *M*. *dirhodum*.

## Figures and Tables

**Figure 1 plants-12-00238-f001:**
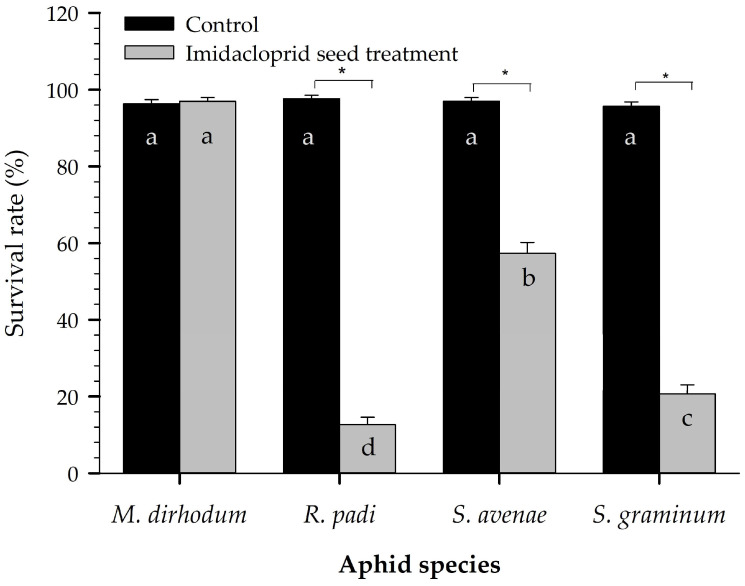
Survival of four wheat aphid species 5 d after they infected wheat on the thirtieth day after germination. “*” above the two bars for a given aphid species indicates that the difference between the control and treatment was significant (*p* < 0.05). The same letters on the control bars indicate that there were no significant differences by paired bootstrap test. Bars showing means of imidacloprid seed treatments with different letters indicate significant differences by paired bootstrap test.

**Figure 2 plants-12-00238-f002:**
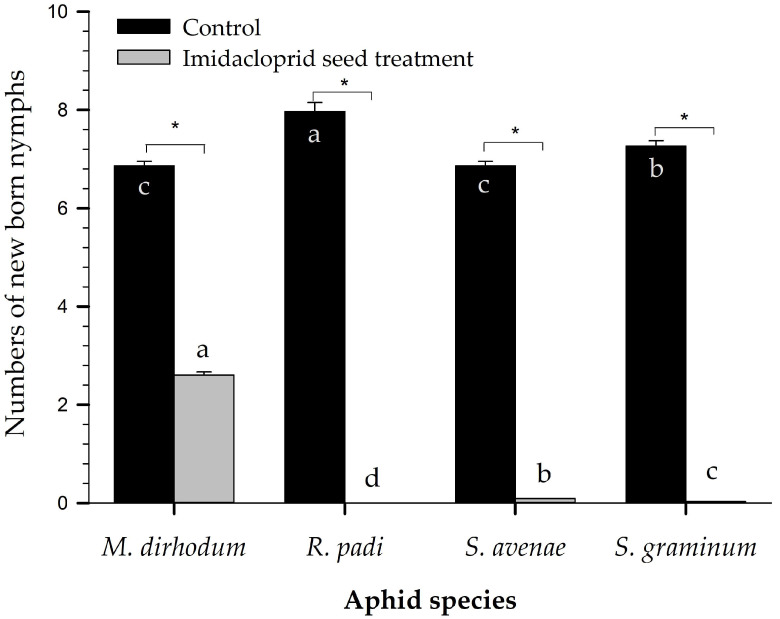
Fecundity of four wheat aphid species 5 d after they infected wheat on the thirtieth day after germination. “*” above the two bars for a given aphid species indicates that the difference between the control and treatment was significant (*p* < 0.05). Means of the same set with different letters were significantly different by paired bootstrap tests.

**Figure 3 plants-12-00238-f003:**
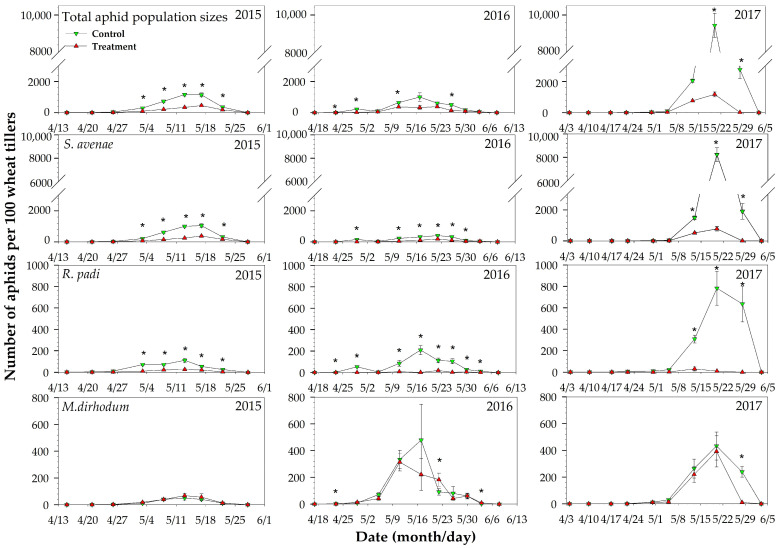
Population dynamics of three aphid species on imidacloprid-treated and control wheat plots in 2015–2017. “*” above the given date indicates a significant difference between the control and imidacloprid seed treatments (*p <* 0.05).

## Data Availability

Not applicable.

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
