# Peer review of "Efficacy of Imidacloprid Seed Treatments against Four Wheat Aphids under Laboratory and Field Conditions"

_plants, 2023, doi:10.3390/plants12020238_

Round 1
Reviewer 1 Report
-Changes needed throughout - highlighted in pdf. It is difficult to read in this current state
-Would advise an additional person reviews writing before resubmission, as grammar issues throughout – I have highlighted some but not all
-Figure 3 needs the same y-axis throughout or you cannot readily compare numbers and is misleading.
-As numbers differed in control plots between years a discussion on the weather/temp would have been useful to understand the fluctuations seen
-Could include discussion on whether different clonal lineages would respond differently to pesticides.
-The effects of imidaclopird has been tested on some aphid species already – should include this in discussion. Some examples include;
Li, W., Lu, Z., Li, L., Yu, Y., Dong, S., Men, X. and Ye, B., 2018. Sublethal effects of imidacloprid on the performance of the bird cherry-oat aphid Rhopalosiphum padi. PLoS One, 13(9), p.e0204097.
Atta, B., Rizwan, M., Sabir, A.M., Gogi, M.D., Farooq, M.A. and Jamal, A., 2021. Lethal and sublethal effects of clothianidin, imidacloprid and sulfoxaflor on the wheat aphid, Schizaphis graminum (Hemiptera: Aphididae) and its coccinellid predator, Coccinella septempunctata. International Journal of Tropical Insect Science, 41(1), pp.345-358.
Kirkland, L.S., Pirtle, E.I. and Umina, P.A., 2018. Responses of the Russian wheat aphid (Diuraphis noxia) and bird cherry oat aphid (Rhopalosiphum padi) to insecticide seed treatments in wheat. Crop and Pasture Science, 69(10), pp.966-973.
- While you refer to Gong et al 2021, I would like to see more discussion of this as it is very relevant for your findings
Gong, P.P.; Chen, D.F; Wang C., Li, M.Y.; Li, X.A.; Zhang, Y.H.; Li, X.R.; Zhu X. Susceptibility of Four Species 405 of Aphids in Wheat to Seven Insecticides and Its Relationship to Detoxifying Enzymes. Front. Physiol. 2021, 406 11:623612.
-Should mention somewhere in the paper other than just the M&M that a recommended dose was used. Lab experiments could benefit from testings number of doses

Author Response
Dear reviewer,
Thank you very much for your elaborative revision to this manuscript. We revised the manuscript according to your suggestions or marks. In addition, this manuscript has undergone English language editing by MDPI and we have revised again.

Reviewer 2 Report
1. General Comments
The paper is very interesting. The work presents a practical problem, characterizes it very well and proposes a good management. In addition, it encourages other research teams to test the effect of pesticides in sibling species, as they can exhibit different behaviours to the same phytopharmaceutical product. As a scientific reader, I find the paper very appealing, motivating, and understandable.
2. Section by section
2.1. Introduction:
The introduction is very easy to read, very comprehensible and has a lot of references to consolidate the affirmations made.
2.2. Material and Methods:
Material and Methods are, from my point the view, well conducted. The statistical methods are adequate, and the sample used is well characterized and representative. In case of interest/need, the description of methods used allows to replicate the assay.
2.3. Results:
Results are well presented; graphic component is good, and the presentation is easy to understand.
2.4. Discussion:
Discussion is well conducted and interesting to read, it is well supported in bibliographic references and clearly explains the observed results.
3. Other comments
I appreciated the article, it was very pleasant to read. It is a source of reliable information, with practical application.
In line 74, I recommend adding the year of publication after the author’s name.
Author Response
Dear reviewer,
Thank you very much for your appreciation.We revised the manuscript according to your suggestions and other reviewers. In addition, this manuscript has undergone English language editing by MDPI and we have revised again.

Reviewer 3 Report
The research methodology is very general, poorly described and not understood. In this form, it is impossible to evaluate the discussion of the results of the research and discussion. I am asking for a correction.
Author Response
Dear reviewer,
Thank you very much for your suggestions to this manuscript. we have revised extensively according the suggestion from all reviewers. This manuscript has undergone English language editing by MDPI and we are very appreciate your evaluation and hope you could give more correction suggestions.
